Response of arbuscular mycorrhizal fungal community in soil and roots to grazing differs in a wetland on the Qinghai-Tibet plateau

Li Zhong-Feng 1 2
Lü Peng-Peng 1 2
Wang Yong-Long 1 2
Yao Hui 1 2
Maitra Pulak 1 2
Sun Xiang 1
Zheng Yong 1
Guo Liang-Dong guold@im.ac.cn 1 2
1 State Key Laboratory of Mycology, Institute of Microbiology, Chinese Academy of Sciences , Beijing , China
2 College of Life Sciences, University of Chinese Academy of Sciences , Beijing , China
Sobral Mar
Electronic publication date: 2020 Jun 19
Publication date: 2020
Volume: 8
Electronic Location ID: e9375
Received 2020 Jan 14; Accepted 2020 May 27
Copyright: ©2020 Li et al.
Copyright year: 2020
Copyright holder: Li et al.
License: This is an open access article distributed under the terms of the Creative Commons Attribution License, which permits unrestricted use, distribution, reproduction and adaptation in any medium and for any purpose provided that it is properly attributed. For attribution, the original author(s), title, publication source (PeerJ) and either DOI or URL of the article must be cited.
License URL: https://creativecommons.org/licenses/by/4.0/

Keywords: Arbuscular mycorrhizal fungal abundance, Community, Diversity, Grazing, Wetland

Funding: National Natural Science Foundation of China 91751113 This study was supported financially by the National Natural Science Foundation of China (no. 91751113). The funders had no role in study design, data collection and analysis, decision to publish, or preparation of the manuscript.

==============================
Grazing as one of the most important disturbances affects the abundance, diversity and community composition of arbuscular mycorrhizal (AM) fungi in ecosystems, but the AM fungi in response to grazing in wetland ecosystems remain poorly documented. Here, we examined AM fungi in roots and soil in grazing and non-grazing plots in Zoige wetland on the Qinghai-Tibet plateau. Grazing significantly increased AM fungal spore density and glomalin-related soil proteins, but had no significant effect on the extra radical hyphal density of AM fungi. While AM fungal richness and community composition differed between roots and soil, grazing was found to influence only the community composition in soil. This study shows that moderate grazing can increase the biomass of AM fungi and soil carbon sequestration, and maintain the AM fungal diversity in the wetland ecosystem. This finding may enhance our understanding of the AM fungi in response to grazing in the wetland on the Qinghai-Tibet plateau.

Introduction

Wetlands cover about 6% of the land surface on the earth and have high species diversity, including many endemic species (Junk et al., 2013). In China, wetlands account for 7% of the wetland on the world (Junk et al., 2013) and have about 225 families, 815 genera and 2,276 species of higher plants (Yan & Zhang, 2005). Wetlands provide important ecological functions in water resource conservation and quality purification, climate regulation, substance circulation and regional ecological balance maintenance (Green et al., 2017). Moreover, as an important carbon (C) pool, wetlands can reduce the impact of increased greenhouse gases on global climate change (Frolking et al., 2011). However, wetland ecosystems have suffered severe degradations in recent decades due to global warming, intense resource exploitation, changes in hydrology and human disturbance (Xiang et al., 2009; Junk et al., 2013). For example, over-grazing as one of the most important human activities, has affected biodiversity, productivity, community stability and soil C cycling in wetlands (Hoffmann et al., 2016; Zhou et al., 2017).

Arbuscular mycorrhizal (AM) fungi, as one of the key components of soil microorganisms, form symbiotic associations with most terrestrial plant species (Smith & Read, 2008). In the AM associations, plants provide C source for the growth and function of fungi, thereby affecting the community of AM fungi (Smith & Read, 2008). In return, AM fungi can increase the nutrient and water absorption of host plants through formation of underground fungal hyphal networks, then affecting plant community and productivity (Van der Heijden, Bardgett & Van Straalen, 2008). Furthermore, AM fungal hyphal and spores produce glomalin-related soil protein (GRSP) which can stably exist in soil and play an important role in soil C pool (Godbold et al., 2006). In addition, AM fungi can improve plants to tolerate grazing and other stresses from the environment (Bennett & Bever, 2007). Thus, revealing the AM fungi in response to grazing is of great importance for understanding the diversity maintenance and community stability of plants in ecosystems, especially in wetland ecosystems.

Previous studies have demonstrated that the effect of grazing on AM fungi is depended on grazing intensity (García & Mendoza, 2012; Kusakabe et al., 2018; Yang et al., 2019). For instance, the light and moderate grazing intensity positively influenced AM fungal spore density in grasslands in Jilin province, China (Ba et al., 2012) and in British Columbia, Canada (Van der Heyde et al., 2017). In contrast, over-grazing negatively affected AM fungal spore density in a semi-arid grassland in China (Su & Guo, 2007). Moderate grazing intensity did not influence AM fungal extra radical hyphal density in an alpine meadow in China (Yang et al., 2013), but high grazing pressure negatively affected the extra radical hyphal density of AM fungi in a semi-arid grassland in China (Ren et al., 2018). Besides, moderate grazing had a neutral effect on AM fungal richness in a meadow in China (Ba et al., 2012). Moderate grazing significantly affected the community composition of AM fungi in soil and roots in grassland ecosystems (Bai et al., 2013; Yang et al., 2013; Kusakabe et al., 2018). In contrast, others found that moderate grazing did not influence the community composition of AM fungi in roots in alpine meadow (Jiang et al., 2018) and in soil in mountain grassland (Van der Heyde et al., 2017) ecosystems. However, previous studies have mainly focused on semi-arid, arid, alpine and mountain grassland ecosystems. So far, we know little about how grazing affects AM fungi in wetland ecosystems.

Zoige wetland is a typical representative of the alpine wetland ecosystem on the Qinghai-Tibet plateau in China, and has high plant species diversity and an important C sink function (Guo et al., 2013). However, Zoige wetland has suffered severe ecosystem degradations since the 1970s, due to global warming, low precipitation and human disturbance, such as ditching for grassland enlargement, peat exploitation and over-grazing (Xiang et al., 2009; Guo et al., 2013). Previous studies have mainly focused on plant diversity, microbial community (archaea group), and ecosystem conservation and restoration in the Zoige Wetland (Wang, Bao & Yan, 2002; Xiang et al., 2009). However, the grazing effect on AM fungi has never been studied.

Overall, the effects of intense grazing on mycorrhizal symbiosis may lead to a shortage on C from the plant to the fungi, what has been related to fungal response with a different production of glomalin and spores and extra radical hyphae of fungal biomass (Hammer & Rillig, 2011; Ba et al., 2012; Van der Heyde et al., 2017). Besides, grazing may cause changes in AM fungal richness and community composition in plant roots and surrounding soil (Bai et al., 2013; Yang et al., 2013). However, these responses are environmentally dependent and habitat-sensitive and may be influenced by soil characteristics and dynamics (Ren et al., 2018; Yang et al., 2019). This study represents the first one to analyze AM fungal responses to grazing in wetland ecosystems.

In order to reveal the AM fungi in response to grazing in wetland ecosystem, we established non-grazing (natural) and moderate grazing plots in Zoige wetland on the Qinghai-Tibet plateau. AM fungal spore density, extra radical hyphal density and GRSP content were examined in grazing and non-grazing plots. We examined the communities of AM fungi in roots and soil by Illumina MiSeq sequencing of 18S rDNA region. We aimed to explore the effect of moderate grazing on AM fungal spore density, extra radical hyphal density, GRSP content, richness and community composition in the Zoige wetland.

Materials and Methods

Study site and sampling

The study was carried out in the centre of Zoige Swamp in the Zoige National Nature Reserve on the Qinghai-Tibet plateau (33° 25′−34° 80′N, 102° 29′−102° 59′E, 16,671 ha, 3,365 m above sea level). The site has a plateau cold temperate humid monsoon climate, with a mean annual temperature (MAT) of 1.1 °C, and a mean annual precipitation (MAP) of 660 mm (Wang, Bao & Yan, 2002). The site begins to freeze in late September and is completely thawed in mid-May (Wang, Bao & Yan, 2002). The abundant plant species are Blysmus sinocompressus, Potentilla anserina, Carex enervis, Caltha scaposa, Elymus nutans and Leontopodium wilsonii in the site (Wang, Bao & Yan, 2002).

The Sichuan Zoige Wetland National Nature Reserve Authority approved the collection of soil and root samples in the Zoige Wetland National Nature Reserve. We established 20 plots (each 1 m ×1 m), > 20 m away from each other, in non-grazing (natural grass) and grazing area, respectively (Fig. S1). The average species number and height of vegetation were 7.8 ± 0.495 (mean ± SE) and ca. 31 cm in the non-grazing plots and 5.4 ± 0.255 and ca. seven cm in the grazing plots. This site was mainly grazed by yaks from June to September, as the growing season is from May to August. The grazing intensity (ca. 1.8 yak/ha) in this study site was described as moderate, as previous study showed that there are three different grazing intensities by yaks (light: 1.2 yaks/ha, moderate: 2.0 yaks/ha, and heavy: 2.9 yaks/ha) in alpine meadow on the eastern Tibetan Plateau (Gao et al., 2007). In July 2018, with the best vegetation growth stage, we randomly collected five soil cores (three cm in diameter; 15 cm in depth; ca. 300 g) and mixed into one composite sample from each plot. A total of 40 samples were obtained, packed in an ice box and transported to our laboratory. Soil samples were sieved (1-mm sieve) to remove debris and roots. Subsoil samples were kept at −80 °C until the extraction of fungal hyphae and DNA, and the remaining subsoil samples were air dried and kept at 10 °C until the analysis of AM fungal spore density, GRSP content and soil properties. We manually collected the mixed roots (< two mm in diameter) from each sieved root sample, washed with sterilized deionized water and kept at −80 °C until DNA extraction.

Soil property analysis

We weighed a certain amount of soil from each sample before and after drying for 24 h in an oven at 105 °C, then calculated the percentage of soil moisture. Soil pH was measured at a ratio of 1:2.5 (w/v, soil: water) with a glass electrode (Thermo Orion T20, Columbia, USA). Soil total nitrogen (N) and C were determined by CHNOS Elemental Analyser (Vario EL III Elementar Analysensysteme GmbH, Germany). Soil total phosphorus (P) was extracted using the HClO4-H2SO4 digestion method and determined with a spectrophotometer (UV-2550, Shimadzu, Japan).

EE-GRSP and T-GRSP

Total GRSP (T-GRSP) and easily extracted GRSP (EE-GRSP) were measured according to the method of Janos, Garamszegi & Beltran (2008). We extracted EE-GRSP from 0.1 g air dried soil using sodium citrate buffer (8 mL, 0.02 M, pH 7.0) at 121 °C for 90 min in an autoclave (Yamato SQ810C, China). We repeatedly extracted T-GRSP from 0.1 g air dried soil using sodium citrate buffer (8 mL, 0.05 M, pH 8.0) at 121 °C for 90 min until no obvious color in the supernatant was observed. Supernatants were separated by centrifugation at 6000 g for 15 min to remove the soil particles and saved in a plastic tube (4 °C). Then 0.5 mL of supernatant of EE-GRSP and T-GRSP was stained with 5 mL of Coomassie Brilliant Blue G-250 and was read in a micro-plate reader (Biotek Synergy H4, Winooski, VT, USA) at 595 nm. The bovine serum albumin was used as a standard solution with Coomassie Brilliant Blue method and a standard curve was drawn to determine the content of EE-GRSP and T-GRSP.

AM fungal extra radical hyphal density and spore density

We extracted fungal hyphae from soil according to the membrane filter method (Rillig, Field & Allen, 1999). In total, 4.0 g of frozen soil from each sample was mixed with 12 mL sodium hexametaphosphate (35 g L−1) and 100 mL distilled deionized water in a flask, and then blended for 30 s, settled for 30 min and sieved (38-µm sieve). The fungal hyphae on the sieve were washed into a flask with 200 mL distilled water, and then 2 mL aliquot was filtered through a 25-µm Millipore filter. The fungal hyphae on the filter were stained with 1% acid fuchsine and distinguished into AM and non-AM fungi on the basis of morphological characteristics and staining color (Miller, Jastrow & Reinhardt, 1995). We measured the hyphal length of AM fungi according to the grid-line intersect method (Tennant, 1975). We extracted AM fungal spores from 20 g air dried soil from each sample according to the wet-sieving and decanting method (Daniels & Skipper, 1982) and counted the spore numbers under 40 × magnification (Nikon 80i, Japan).

DNA extraction, PCR and Illumina Miseq sequencing

We extracted DNA from 0.2 g frozen roots and soil using the PowerSoil® DNA isolation kit (MOBIO Laboratories, Inc., Carlsbad, USA) in accordance with the manufacturer’s instructions, and measured the DNA concentration using a NanoDrop 1000 Spectrophotometer (Thermo Scientific, Wilmington, USA). We amplified the fungal 18S rDNA region using a two-step PCR procedure. The first PCR using primers AML2 (Lee, Lee & Young, 2008) and GeoA2 (Schwarzott & Schüßler, 2001) was conducted in a final 25 µL reaction mixture, including ca. 10 ng of template DNA, 0.75 µM of each primer, 250 µM of each dNTP, 0.5 U KOD-plus-Neo polymerase (Toyobo, Tokyo, Japan), 1.5 mM MgSO4, and 2.5 µL 10 × buffer. The thermal cycling conditions were performed as follows: an initial denaturation at 95 °C for 5 min, 30 cycles for denaturation at 94 °C for 1 min, annealing at 58 °C for 50 s and extension at 68 °C for 1 min, and a final extension at 68 °C for 10 min. The products of the first amplification were diluted 100 times, and 1 µL of the diluted DNA template was used for the second amplification. The thermal cycling conditions for the second amplification were the same as first amplification, except that the primers NS31 (Simon, Lalonde & Bruns, 1992) and AMDGR (Sato et al., 2005) linked with 12-base barcode sequences were used. The size of amplified fragment was about 300 base pairs (bp). We purified the PCR products using a PCR Product Gel Purification Kit (Omega Bio-Tek, USA), and pooled the purified PCR products with the same amount (100 ng) from each sample and adjusted the concentration to 10 ng µL−1. We constructed a sequencing library by addition of an Illumina sequencing adaptor (5′-GATCGGAAGAGCACACGTCTGAACTCCAGTCACATCACGATCT- CGTATGCCGTCTTCTGCTTG-3′) to the products using the Illumina TruSeq DNA PCR-Free LT Library Prep Kit (Illumina, CA, USA) according to the manufacturer’s instructions. We sequenced the library by an Illumina MiSeq PE 250 platform using the paired-end (2 × 250 bp) option in the Chengdu Institute of Biology, Chinese Academy of Sciences, China.

Bioinformatics analysis

We filtered the raw sequences using Quantitative Insights into Microbial Ecology (QIIME) v.1.7.0 (Caporaso et al., 2010) to eliminate low-quality sequences, such as read length < 200 bp, no valid primer sequence or barcode sequence, containing ambiguous bases, or an average quality score < 20. We checked and deleted the potential chimeras against the MaarjAM database (Öpik et al., 2010) using the ‘chimera.uchime’ command in Mothur version 1.31.2 (Schloss et al., 2009). High quality sequences were subjected to de-replication and de-singleton, and then clustered into operational taxonomic units (OTUs) at a 97% sequence similarity level using the cluster_otus command in USEARCH v8.0 (Edgar, 2013). Using a basic local alignment search tool (BLAST) (Altschul et al., 1990), we selected the most abundant sequence of each OTU and searched against the MaarjAM database and National Center for Biotechnology Information (NCBI) nt database. We identified OTUs as the AM fungi based on the closest BLAST hit annotated as ‘Glomeromycotina’ and E values < e−50. Furthermore, we normalized the sequence number of each sample to the smallest sample size using the ‘sub.sample’ command in Mothur. We have submitted the representative sequence of each AM fungal OTU to the European Molecular Biology Laboratory (EMBL) database (accession no. LR736402-LR736557). The identified AM fungi are shown in Table S1 .

Statistical analysis

We conducted all statistical analyses in R version 3.3.2 (R Development Core Team, 2017). Tukey’s honestly significant difference (HSD) test or Conover’s test was used to examine the significant difference of soil moisture, pH, total N, total C and total P in the grazing and non-grazing plots at P < 0.05. Generalized linear model (GLM) with Poisson error structure and log link function was conducted to evaluate the effect of grazing on AM fungal spore density, extra radical hyphal density and T-GRSP, as these data did not meet normal distribution, while GLM with Gaussian error structure and identity link function was conducted to evaluate the effect of grazing on EE-GRSP, and then Conover’s test was used to examine the significant difference between grazing and non-grazing treatments at P < 0.05 using the post-hoc.kruskal.conover.test function in the PMCMR package (Pohlert, 2014). Meanwhile, GLM with Gaussian error structure and identity link function was used to evaluate the effect of grazing and sample type nested in grazing on the AM fungal OTU richness, and GLM with Gamma error structure and inverse link function was used to evaluate the effect of grazing and sample type nested in grazing on the relative abundance of abundant OTUs (relative abundance > 1%) and orders of AM fungi, as these data did not meet normal distribution, and then Conover’s test was conducted for comparisons between grazing and non-grazing treatments in soil and roots at P <  0.05.

The distance matrices of AM fungal community composition (Hellinger-transformed OTU read data) in roots and soil were established by the Bray–Curtis method (Clarke, Somerfield & Chapman, 2006). Nested permutational multivariate analysis of variance (PerMANOVA) was conducted to examine the effect of grazing and sample type nested within grazing on AM fungal community composition, using the ‘adonis’ function in the vegan with 999 permutations (Oksanen et al., 2013). Besides, PerMANOVA was conducted to examine the effect of grazing on AM fungal community composition in soil and roots, respectively. Redundancy analysis (RDA) was conducted to reveal the significant correlation of AM fungal community composition and soil variables using the Monte Carlo permutation test with 999 permutations.

Results

Soil properties

Soil pH was significantly lower in grazing treatment than in non-grazing treatment (Table 1). Soil moisture, total N, total C and total P were significantly higher in grazing treatment than in non-grazing treatment (Table 1).

Table 1 Soil properties in grazing and non-grazing treatments in this study.

Soil variable	Grazing	Non-grazing	
pH	7.719 ± 0.215 b	7.838 ± 0.067 a	
Moisture (%)	35.09 ± 6.307 a	29.24 ± 4.766 b	
N (g kg−1)	14.92 ± 5.588 a	8.629 ± 3.360 b	
C (g kg−1)	202.5 ± 76.46 a	111.9 ± 36.70 b	
P (g kg−1)	1.232 ± 0.242 a	1.036 ± 0.084 b	
Notes.

Data (means ± SD) with different letters in the same row are significantly different at P < 0.05, as indicated by Tukey’s HSD test or Conover’s test.

N soil total nitrogen

C soil total carbon

P soil total phosphorus

EE-GRSP and T-GRSP contents

The EE-GRSP content was 28.96  ± 3.73 µg g−1 (mean  ± SE) and 25.71  ± 2.26 µg g−1 in grazing and non-grazing treatments, respectively. The T-GRSP content was 96.7  ± 18.82 µg g−1 and 71.87   ± 12.87 µg g−1 in grazing and non-grazing treatments, respectively. GLM showed that grazing significantly influenced EE-GRSP (P = 0.002; Table 2) and T-GRSP (P < 0.001; Table 2). For example, EE-GRSP and T-GRSP contents were significantly lower in non-grazing than in grazing treatments (Figs. 1A and 1B).

Table 2 General linear model (GLM) showing the effect of grazing on easily extracted glomalin-related soil protein (EE-GRSP), total extracted GRSP (T-GRSP), spore density and extra radical hyphal (ERH) density of arbuscular mycorrhizal fungi.

Variable	Estimate	SE	t/z-value	P-value	
EE-GRSP	−3.2506	0.975	−3.334	0.002	
T-GRSP	−0.29705	0.03482	−8.531	<0.001	
Spore density	−0.54395	0.07252	−7.501	<0.001	
ERH density	−0.2562	0.1693	−1.514	0.13	

Figure 1 Easily extracted glomalin-related soil protein (EE-GRSP, A), total extracted GRSP (T-GRSP, B), spore density (C) and extra radical hyphal (ERH) density (D) of arbuscular mycorrhizal fungi in grazing and non-grazing treatments.

Data are means ± SE (n = 20). Bars with different letters denote significant difference in grazing and non-grazing treatments according to Conover’s test at P < 0.05.

AM fungal spore density and extra radical hyphal density

The spore density of AM fungi was 25.89  ± 12.17 g−1 (mean  ± SE) and 15.03 ± 5.88 g−1 in grazing and non-grazing treatments, respectively. The extra radical hyphal density of AM fungi was 4.00  ± 2.51 m g−1 and 3.10  ± 1.56 m g−1 in grazing and non-grazing treatments, respectively. GLM revealed that grazing significantly affected AM fungal spore density (P = 0.001; Table 2) but not extra radical hyphal density (P = 0.130; Table 2). For example, the spore density of AM fungi was significantly lower in non-grazing than in grazing treatments (Fig. 1C). However, AM fungal extra radical hyphal density was not significantly different in non-grazing and grazing treatments (Fig. 1D).

Identification of AM fungi

In total, 3,205,557 high-quality sequences were filtered from 3,335,816 raw sequences and clustered into 882 OTUs at a 97% sequence similarity level. Among 882 OTUs, 156 (2,919,706 sequences) belonged to AM fungi. As the sequence number of AM fungi varied from 20,408 to 48,572 in the 80 samples, the number of sequence was normalized to 20,408. The normalized dataset contained 156 AM fungal OTUs (1,632,640 sequences). Of the 156 AM fungal OTUs obtained, 154 were from soil, 152 from roots, and 150 shared both soil and roots. Among 156 AM fungal OTUs, 153 were detected from more than three samples (frequency ≥ 3.75%) (Fig. S2A). Furthermore, the 21 abundant AM fungal OTUs (relative abundance > 1%) occupied 83.85% of the total sequences (Fig. S2B). Among 156 AM fungal OTUs, 109 were identified to Glomerales (79.52% of sequences), 22 to Diversisporales (10.84%), 21 to Archaeosporales (8.75%), and 4 to Paraglomerales (0.89%). In addition, the rarefaction curves indicated that the sample numbers were sufficient to detect the most AM fungi in this study (Fig. S3).

AM fungal OTU richness

AM fungal OTU richness in grazing and non-grazing treatments was 123.70  ± 2.96 (mean  ± SE) and 122.85  ± 3.01 in soil, and 117.55  ± 2.26 and 118.00  ± 4.38 in roots, respectively. GLM revealed that AM fungal OTU richness was influenced by sample type (root and soil; P < 0.001; Table S2), but not by grazing (P = 0.662; Table S2). For example, AM fungal OTU richness was significantly lower in roots than in soil in both grazing and non-grazing treatments (Fig. 2). However, AM fungal OTU richness was not significantly different between grazing and non-grazing treatments in roots and soil, respectively (Fig. 2).

Figure 2 The operational taxonomic unit (OTU) richness of arbuscular mycorrhizal fungi in soil and roots in grazing and non-grazing treatments.

General linear model (GLM) showing the effect of grazing and sample type (soil and root) on the OTU richness. Data are means ±  SE (n = 20). Bars with different letters denote significant difference in grazing and non-grazing treatments according to Conover’s test at P < 0.05. G, grazing; NG, non-grazing; ST, sample type.

AM fungal community

GLM revealed that grazing had significant effect on the relative abundance of abundant AM fungal OTU12 and OTU25 (Glomerales), and sample type had significant effect on the relative abundance of abundant AM fungal OTU4, OTU5, OTU7, OTU12, OTU14, OTU18, OTU25 and OTU141 (Glomerales), OTU8 and OTU17 (Diversisporales) and OTU23 (Archaeosporales) (Fig. 3; Tables S3 ; S4).

GLM revealed that sample type significantly influenced the relative abundance of Glomerales, Diversisporales and Archaeosporales, and grazing significantly affected the relative abundance of Diversisporales (Fig. 4; Table S5). The relative abundance of Glomerales was significantly lower in soil than in roots; by contrast, the relative abundance of Diversisporales and Archaeosporales was significantly lower in roots than in soil, regardless of non-grazing and grazing treatments (Fig. 4; Table S5). Besides, the relative abundance of Diversisporales was significantly lower in grazing treatment than in non-grazing treatment (Fig. 4; Table S5).

Figure 3 Relative abundance of arbuscular mycorrhizal (AM) fungal operational taxonomic units (OTUs) in soil and roots in grazing and non-grazing treatments.

General linear model (GLM) showing the effect of grazing and sample type (soil and root) on the relative abundance of AM fungal OTUs (ns; P ≥ 0.05, ∗ P < 0.05, ∗∗ P < 0.01, ∗∗∗ P < 0.001). The rare AM fungal OTUs (< 1% of total AM fungal reads) and abundant AM fungal OTUs (> 1% of total AM fungal reads) that was not significantly affected by grazing and sample type were all assigned to “Others”. SN, soil non-grazing; SG, soil grazing; RN, root non-grazing; RG, root grazing; G, grazing; NG, non-grazing; ST, sample type.

Figure 4 Relative abundance of arbuscular mycorrhizal (AM) fungi at the order level in soil and roots in grazing and non-grazing treatments.

General linear model (GLM) showing the effect of grazing and sample type (soil and root) on the relative abundance of AM fungal orders (ns; ** P < 0.01, *** P < 0.001). Different letters are significantly different at P < 0.05, as indicated by Conover’s test. SN, soil non-grazing; SG, soil grazing; RN, root non-grazing; RG, root grazing; G, grazing; NG, non-grazing; ST, sample type.

The PerMANOVA demonstrated that the community composition of AM fungi was significantly influenced by sample type (soil and root; F = 7.2836, R2 = 0.157, P = 0.001; Table S6) and grazing (F = 2.339, R2 = 0.025, P = 0.012; Table S6). Furthermore, the community composition of AM fungi was significantly influenced by grazing in soil (F = 2.639, R2 = 0.055, P = 0.001; Table S7), but not in roots (F = 0.998, R2 = 0.025, P = 0.419; Table S8). Furthermore, RDA showed that the community composition of AM fungi in soil and roots was significantly correlated with soil pH, moisture, total C, total N and total P (Fig. 5).

Figure 5 Redundancy analysis (RDA) biplots showing arbuscular mycorrhizal fungal community composition in soil and roots (A), soil (B) and roots (C).

Significant soil variables were presented as vectors on the RDA biplot graphs using the ‘envfit’ (based on 999 permutations) at P < 0.05. SN, soil non-grazing; SG, soil grazing; RN, root non-grazing; RG, root grazing; N, soil total nitrogen; C, soil total carbon; P, soil total phosphorus.

Discussion

We found that grazing had positive effect on AM fungal spore density, EE-GRSP and T-GRSP, in consistent with some previous studies (Hammer & Rillig, 2011; Yang et al., 2013; Van der Heyde et al., 2017). Previous findings suggest that moderate removal of aboveground biomass may increase the allocation of C to the roots and exudation from roots to soil (Eom, Wilson & Hartnett, 2001; Soka & Ritchie, 2018), which could be beneficial for the sporulation of AM fungi (Ba et al., 2012; Van der Heyde et al., 2017). Furthermore, since about 80% of GRSP is produced by the AM fungi, moderate grazing increased AM fungal spore density, resulting in increasing GRSP content in soil (Driver, Holben & Rillig, 2005). This suggests that grazing plays an important role in soil C pool in the wetland ecosystem on the Qinghai-Tibet Plateau (Gao et al., 2007). However, grazing did not significantly influence AM fungal extra radical hyphal density, as reported in a previous study (García & Mendoza, 2012). Although moderate grazing may increase C allocation to the roots, this increase may be ephemeral (Van der Heyde et al., 2019) and not be sufficient to promote the growth of AM fungal hyphae.

AM fungal richness was significantly lower in roots than in soil, as previous studies reported in alpine and meadow ecosystems (Hempel, Renker & Buscot, 2007; Liu et al., 2012; Yang et al., 2013). This may be that the currently and formerly active propagules of AM fungi could remain in soil; by contrast, only currently active AM fungi could occur in the roots (Liu et al., 2009; Martínez-García et al., 2011). However, we found that grazing did not significantly influence AM fungal richness in roots and soil. Similarly, a previous study showed that moderate grazing could maintain the AM fungal diversity (Dudinszky et al., 2019). In general, AM fungi have low specificity (Smith & Read, 2008), thus AM fungal richness may not be influenced by the low plant species diversity caused by moderate grazing, as some studies found that AM fungal richness was not related to plant species diversity (Wolf et al., 2003).

The community composition of AM fungi significantly differed between roots and soil in this study, as previous studies reported in grassland (Yang et al., 2013), farmland (Liu et al., 2016) and temperate (Saks et al., 2014) and subtropical forest (Maitra et al., 2019) ecosystems. This may be explained by the difference in AM fungal abundance in roots and soil (Hempel, Renker & Buscot, 2007; Maitra et al., 2019). Indeed, our result found that some AM fungi were abundant in roots and soil, respectively. In addition, AM fungal phenology may produce different communities in soil and roots (Liu et al., 2012).

Grazing significantly affected the AM fungal community composition in soil, in consistent with some previous studies reported in desert steppe and grassland ecosystems (Murray, Frank & Gehring, 2010; Bai et al., 2013). Grazing may influence the AM fungal community composition by changing soil properties through animal trampling and fecal deposition (McNaughton, Banyikwa & McNaughton, 1997; Yang et al., 2019). For example, animal trampling may make the soil tight and alter soil bulk density (Kauffman, Thorpe & Brookshire, 2004; Byrnes et al., 2018), thereby influencing AM fungal community (Yang et al., 2018). Moreover, dung and urine produced by animals, as soil fertilization, may decrease soil pH and increase soil nutrients as shown in this and previous studies (McNaughton, Banyikwa & McNaughton, 1997; Kohler et al., 2005), thus altering AM fungal community composition. Indeed, our result showed that the community composition of AM fungi was significantly related to soil pH, moisture, total C, total N and total P, as previous studies reported in semi-arid, alpine and temperate grassland and subtropical forest ecosystems (Zheng et al., 2014; Gao et al., 2016; Zhang et al., 2016; Goldmann et al., 2019; Maitra et al., 2019). However, the AM fungal community composition in roots was not significantly influenced by grazing, as previous studies reported in semi-arid and alpine grassland ecosystems (González et al., 2018; Jiang et al., 2018). It is possible that moderate grazing does not much change the allocation of carbohydrates to roots, thereby without altering AM fungal community. Furthermore, although grazing may alter the AM fungal function, it does not necessarily alter the community in roots (González et al., 2018).

Conclusions

In conclusion, we examined the AM fungi in response to grazing in the Zoige wetland on the Qinghai-Tibet plateau for the first time. AM fungal spore density and GRSP content positively responded to grazing. The extra radical hyphal density and OTU richness of AM fungi had neutral response to grazing. The community composition of AM fungi was significantly influenced by grazing in soil but not in roots. These findings suggest that moderate grazing can increase the biomass of AM fungi and soil C sequestration, and maintain the AM fungal diversity in the wetland ecosystem on the Qinghai-Tibet Plateau. Future studies can focus on measuring C flux between AM fungi and host to fully understand the role of grazing on AM fungal function in wetland ecosystems.

Supplemental Information

Supplemental Information 1 Photos showing the study site in Zoige wetland ecosystem

(a) vegetation in grazing plots, (b) vegetation in non-grazing plots.

Click here for additional data file.

Supplemental Information 2 Rank of the arbuscular mycorrhizal (AM) fungal operational taxonomic units (OTUs) by (a) frequency and (b) abundance

Click here for additional data file.

Supplemental Information 3 Rarefaction curves for the observed arbuscular mycorrhizal fungal operational taxonomic units (OTUs) in grazing and non-grazing treatments of soil and roots

SN, soil non-grazing; SG, soil grazing; RN, root non-grazing; RG, root grazing.

Click here for additional data file.

Supplemental Information 4 Molecular identification of arbuscular mycorrhizal fungi at the 97% sequence identity level

Click here for additional data file.

Supplemental Information 5 General linear model (GLM) showing the effect of grazing and sample type (soil and root) on the operational taxonomic unit richness of arbuscular mycorrhizal fungi

Click here for additional data file.

Supplemental Information 6 The effect of grazing and sample type (soil and root) on the relative abundance of arbuscular mycorrhizal fungal operational taxonomic units

G, grazing; NG, non-grazing; ST, sample type.

Click here for additional data file.

Supplemental Information 7 Comparison of the relative abundance of abundant arbuscular mycorrhizal (AM) fungal operational taxonomic units (OTUs, relative abundance ¿1%) in soil and roots in grazing and non-grazing treatments

Click here for additional data file.

Supplemental Information 8 General linear model (GLM) showing the effect of grazing and sample type (soil and root) on the relative abundance of arbuscular mycorrhizal fungal orders

G, grazing; NG, non-grazing; ST, sample type.

Click here for additional data file.

Supplemental Information 9 Nested permutational multivariate analysis of variance (PerMANOVA) testing the effect of grazing and sample type (root and soil) on arbuscular mycorrhizal fungal community composition

Click here for additional data file.

Supplemental Information 10 Permutational multivariate analysis of variance (PerMANOVA) testing the effect of grazing on arbuscular mycorrhizal fungal community composition in soil

Click here for additional data file.

Supplemental Information 11 Permutational multivariate analysis of variance (PerMANOVA) testing the effect of grazing on arbuscular mycorrhizal fungal community composition in roots

Click here for additional data file.

Supplemental Information 12 The raw data including soil properties, AM fungal spore density and extra radical hyphal density, total glomalin-related soil protein and easily extracted glomalin-related soil protein

Click here for additional data file.

Supplemental Information 13 Sequencing data: Soil non-grazing 1-5

Click here for additional data file.

Supplemental Information 14 Sequencing data: Soil non-grazing 6-10

Click here for additional data file.

Supplemental Information 15 Sequencing data: Soil non-grazing 11-15

Click here for additional data file.

Supplemental Information 16 Sequencing data: Soil non-grazing 16-20

Click here for additional data file.

Supplemental Information 17 Sequencing data: Soil grazing 1-5

Click here for additional data file.

Supplemental Information 18 Sequencing data: Soil grazing 6-10

Click here for additional data file.

Supplemental Information 19 Sequencing data: Soil grazing 11-15

Click here for additional data file.

Supplemental Information 20 Sequencing data: Soil grazing 16-20

Click here for additional data file.

Supplemental Information 21 Sequencing data: Root non-grazing 1-5

Click here for additional data file.

Supplemental Information 22 Sequencing data: Root non-grazing 6-10

Click here for additional data file.

Supplemental Information 23 Sequencing data: Root non-grazing 11-15

Click here for additional data file.

Supplemental Information 24 Sequencing data: Root non-grazing 16-20

Click here for additional data file.

Supplemental Information 25 Sequencing data: Root grazing 1-5

Click here for additional data file.

Supplemental Information 26 Sequencing data: Root grazing 6-10

Click here for additional data file.

Supplemental Information 27 Sequencing data: Root grazing 11-15

Click here for additional data file.

Supplemental Information 28 Sequencing data: Root grazing 16-20

Click here for additional data file.

We thank Ga E, Ying Ban and Hongjun Wang from Zoige Wetland National Nature Reserve Authority for help during sampling.

Additional Information and Declarations

Competing Interests

Author Contributions

Field Study Permissions

Data Availability

The authors declare there are no competing interests.

Zhong-Feng Li conceived and designed the experiments, performed the experiments, analyzed the data, prepared figures and/or tables, authored or reviewed drafts of the paper, and approved the final draft.

Peng-Peng Lü analyzed the data, prepared figures and/or tables, and approved the final draft.

Yong-Long Wang performed the experiments, prepared figures and/or tables, and approved the final draft.

Hui Yao analyzed the data, authored or reviewed drafts of the paper, and approved the final draft.

Pulak Maitra performed the experiments, prepared figures and/or tables, and approved the final draft.

Xiang Sun conceived and designed the experiments, performed the experiments, authored or reviewed drafts of the paper, and approved the final draft.

Yong Zheng conceived and designed the experiments, authored or reviewed drafts of the paper, and approved the final draft.

Liang-Dong Guo conceived and designed the experiments, authored or reviewed drafts of the paper, and approved the final draft.

The following information was supplied relating to field study approvals (i.e., approving body and any reference numbers):

The Sichuan Zoige Wetland National Nature Reserve Authority approved the collection of soil and root samples in the Zoige Wetland National Nature Reserve.

The following information was supplied regarding data availability:

Data are available at EMBL: Project PRJEB35257, LR736402– LR736557.

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
