# Peer review of "Response of arbuscular mycorrhizal fungal community in soil and roots to grazing differs in a wetland on the Qinghai-Tibet plateau"

_PeerJ, doi:10.7717/peerj.9375_

## Round 0.1 · original submission · Major Revisions

Dear authors, I will be willing to reconsider your manuscript after the major changes suggested by the three reviewers are implemented.

Best regards

Reviewer 1 ·

Basic reporting

The hypotheses are not well developed (not clearly answered either). Listing what previous studies have reported about AM fungal changes in other systems doesn’t automatically give the basis and the logic development of your own hypotheses. I’d like to see why you hypothesize moderate grazing would increase AM fungal spore density and GRSP content, but not to change the hyphal density, and why moderate grazing would change the community composition of AM fungi but not richness in roots and soil in the Zoige wetland.

The language needs to be improved substantially. Quite many sentences are difficult to understand, for example Lines 37-38, Line 43 (the use of “On the contrary”), Line3 234-235, etc.

Experimental design

According to the description of your sampling strategy, root and soil have to be considered as paired samples in statistical analysis. The paired characteristic of root and soil sample requires root versus soil to be nested in grazing treatment, i.e. nested ANOVAs and nested PerMANOVA should be used.

Validity of the findings

Should be more explicit about whether the two hypotheses are supported or not.

Additional comments

This study examined whether and how grazing would change arbuscular mycorrhizal (AM) fungal spore density, hyphal density and GRSP content in soil and fungal richness and community composition in both roots and soil in a Qinghai-Tibet Plateau wetland. The authors reported that moderate grazing increased AM fungal spore density and GRSP content, but had no effect on hyphal density and fungal OUT richness. AM fungal community composition differed between roots and soil, and was significantly influenced by grazing in soil but not in roots. Overall, the methodology and findings are sound and the manuscript is easy to follow.

My concerns and comments are:
1, The hypotheses are not well developed (not clearly answered either). Listing what previous studies have reported about AM fungal changes in other systems doesn’t automatically give the basis and the logic development of your own hypotheses. I’d like to see why you hypothesize moderate grazing would increase AM fungal spore density and GRSP content, but not to change the hyphal density, and why moderate grazing would change the community composition of AM fungi but not richness in roots and soil in the Zoige wetland.
2, The language needs to be improved substantially. Quite many sentences are difficult to understand, for example Lines 37-38, Line 43 (the use of “On the contrary”), Line3 234-235, etc.
3, According to the description of your sampling strategy, root and soil have to be considered as paired samples in statistical analysis. The paired characteristic of root and soil sample requires root versus soil to be nested in grazing treatment, i.e. nested ANOVAs and nested PerMANOVA should be used.

Line 94: Sources of MAT and MAP?
Lines 102-103: Provide the criteria for defining moderate vs light grazing intensity.
Lines 109-110: but the soil was passed through 1-mm sieve.
Line 114: This is sufficient for the description of soil moisture measurement.
Lines 194-196: Results of soil analyses need to be reported in the section Results.
Line 310: what does “seasonal nature” mean? be explicit.
Lines 261-279: The IDs of these OTUs are meaningless, so it seems quite redundant to list them. What the point to report meaningless OUT IDs here?

Reviewer 2 ·

Basic reporting

The draft is well written and it is a descriptive study of the effect of grazing to the arbuscular mycorrhizal fungi communities and activities in wetlands. The background context is good. The structure is clear and figures are self-explanatory. However, the hypotheses are not well introduced, I miss a bit more background on the meaning of the variables measured in relation to the effect of grazing.

Experimental design

The aim fits the journal and the questions/aims are relevant, although its meaning is not clear in the introduction. the design is fine the statistical analyses should be improve moving from ANOVA to GLM in particular when the data is lof or square-root transformed (see comment to authors for details).

Validity of the findings

I suggest repeating the statistical analyses, as the model assumption might not have met in the current ones. I lacks a deeper interpretation in the context of the paper, so a little speculation might help in the discussion and a more ecological conclusion is needed (instead of a description of results).

Additional comments

The manuscript of dr. Zhong-Feng Li and coworkers deals with an interesting topic, the effect of moderate grazing to the soil properties in relation to arbuscular mycorrhizal fungi (AMF), in terms of soil glomalin content, AMF spore density, extraradical hyphal density and AMF richness and composition in soils and roots. The text is clear and easy to follow and the results seem logical, however, there is relevant information missing about the context how was the grazing areas (density of herbivores and grazing period), and the logic of choosing all of these measurements related to AMF and the timing of sampling. Besides the introduction and discussion goes smoothly but maybe too superficially through the meaning of the measurements and the results, see my main concerns below.
*.- I feel the introduction and discussion are superficial and do not introduce the logic and scientific meaning of measuring soil glomalin, soil spores density and extraradical hyphal density in wetlands. Thus i recommend the authors to expand the reasoning and expectation of measuring these variables in this particular habitat, possibly separating each measure a bit more. This would lead to having a more clear hypothesis and a more deep discussion about them.
*.- Some of the analytical methods sound not completely appropriate. For example the use of ANOVA for variables that are not normally distributed or are discrete (richness) or non-negative (densities), force the use of data transformation. These transformations have been (in the last 10 years) highly unrecommended as they modify the data characteristics sometimes in unpredictable ways. See for example:
O’Hara, R.B. and Kotze, D.J. (2010), Do not log‐transform count data. Methods in Ecology and Evolution, 1: 118-122. doi:10.1111/j.2041-210X.2010.00021.x
Warton, D.I. and Hui, F.K.C. (2011), The arcsine is asinine: the analysis of proportions in ecology. Ecology, 92: 3-10. doi:10.1890/10-0340.1
Besides the transformation step, also the ANOVA does not help in analyses validation as residuals are not estimated. Thus I strongly recommend the authors to move to generalized linear models (using Poisson or negative-binomial for counts and gamma for non-negative continuous and see whether the model is valid) and provide validation with the residuals of the model vs the fitted values and check heteroscedasticity in the residuals. For guidance I recommend:
Mixed E_ects Models and Extensions in Ecology with R
Alain F. Zuur, Elena N. Ieno, Neil J. Walker, Anatoly A. Saveliev, Graham M. Smith
Springer-Verlag, New York, 2009. ISBN 978-0-387-87457-9. 574 pp.
*.- The conclusion just describes the results and not the consequences of the results in the context of our knowledge and specifically in wetlands.

·

Basic reporting

The manuscript is well written, with enough information about the area of research, however I think that the authors should reduce the number of references by choosing the relevant, since there ae more than ten pages of references. This manuscript is well estructured, with figures and raw data shared. The results are clear, however the hypotheses can not be proved and supported with the experimental design choseen by the authors (see next section).

Experimental design

The research question is well defined, however the experimental design do not allowed to prove the hypotheses, since there are no previous knowledge about the arbuscular mycorrhizal fungi communities in the grazing treatment, so it is hard to know if the communities detected in this research are the result of grazing treatment.
I would suggest to change the research question, in order to improve the manuscript findings, consequently the data analysis and statistics analysis must change.

Validity of the findings

Since the experimental desing do not allowed to prove the hypotheses, the conclusion the authors reach are not well supported with the results obtained.

---

## Round 0.2 · Minor Revisions

Dear authors,

Although reviewers are mostly satisfied, reviewer 2 suggests additional improvements to your manuscript. I agree those changes will benefit the manuscript. Please modify accordingly.

Reviewer 2 ·

Basic reporting

The reviewed manuscript is clear, well written, with appropriate references, manuscript structure and with an improved interpretation of the results.

Experimental design

The question section, at the end of the introduction, can be further improved by adding a line explaining the meaning of the selection of the response variables in the text. I think the authors integrate better this information in the discussion, but in the introduction, readers could not get the aim of the paper, and why all these variables are chosen and not others. I tried to be more explicit and suggest a line structure to add, see my comments in the general comments to the authors' section.

Validity of the findings

The findings of this paper are interesting, as this is the first study to my knowledge of grazing effect on AM fungal properties in wetlands. The data and analyses have been fixed and seem to be appropriate. Conclusions are logical and reasonable. Authors could speculate a bit more given all the information provided in the literature and with their own results, identifying it as the potential mechanisms ruling in the conditions given. This speculation is key to the development of further hypothesis in future studies.

Additional comments

The manuscript has improved significantly and most of my previous concerns have been successfully fixed. However, I found still some issues I recommend the authors to deal with before further considerations:
The questions/hypotheses. All this information is more clear and integrated into the discussion. The authors should consider adding a line integrating the meaning of the measurements selected. It is still not completely clear in the introduction why authors have chosen these variables and only readers can understand it when reading the discussion. In other words, some bits of this information needs to go to the introduction. I suggest the authors an integrating line such as:
„Overall, the effects of intense grazing on mycorrhizal symbiosis may lead to a shortage on C from the plant to the fungi, what has been related to fungal response with a different production of glomalin, spores and extraradical fungal biomass (add refs): Besides grazing may cause changes in AM fungal richness and community composition in plant roots and surrounding soil (refs). However, these responses are environmentally dependent and habitat-sensitive and may be influenced by soil characteristics and dynamics (refs). This study represents the first one to analyze AM fungal responses to grazing in Wetland ecosystems.“
The abstract needs a more sound final line, perhaps rephrasing and bringing to the abstract the last new a nice line of the conclusion. Also, I recommend combining a line of the results in the abstract for better information flow. For example replace „AM fungal richness was significantly lower in roots than in soil, but not significantly influenced by grazing. AM fungal community composition was significantly different between roots and soil, and was significantly influenced by grazing in soil but not in roots. “ with something like „While AM fungal richness and community composition differed between roots and soils, grazing was found to influence only the community composition in soils“. Then there is more room to interpret these results in the abstract.

·

Basic reporting

The manuscript is well written and now the number of references is adequate, the results are clear, and the hypotheses has been changed to questions by the authors, so it is better according to work they present.

Experimental design

The authors have now replaced the hypotheses by question to explore the effect of moderate grazing on AM fungal parameters and GRSP content, richness and community composition. The methods are sufficiently described.

Validity of the findings

Now the authors showed a well defined questions, and the conclusions are well structured and are supported by results.

Additional comments

The manuscript are now been improved, the question are answered appropriately, the methods and results are clear, the discussions and conclusion are referred to result reached.

---

## Round 0.3 · accepted · Accept

Thanks for your contribution to PeerJ.